# Sustainability Perspective of Minjiang Estuary Coastal Fisheries Management—Estimation of Fish Richness

Jia-Qiao Wang [1,2,†], Jun Li [1,2,†], Yi-Jia Shih [1,2,*], Liang-Min Huang [1,2], Xin-Ruo Wang [1,2] and Ta-Jen Chu [1,2]

1 Fisheries College, Jimei University, Xiamen 361021, China; skyofstar1@jmu.edu.cn (J.-Q.W.); lijun1982@jmu.edu.cn (J.L.); lmhuang@jmu.edu.cn (L.-M.H.); rouryeah@163.com (X.-R.W.); chutajen@gmail.com (T.-J.C.)
2 Fujian Provincial Key Laboratory of Marine Fishery Resources and Eco-Environment, Xiamen 361021, China
* Correspondence: eja0313@gmail.com
† These authors contributed equally to this work.

**Abstract:** Species richness is the most basic concept of diversity and is crucial to biodiversity conservation and sustainable fisheries. To understand the fish species richness of the Minjiang Estuary and its adjacent waters, eight documents and surveyed data were collected and compared from 1990–2021. To obtain suitable analysis data, the content of the data was compared and evaluated. Explore the suitability of data based on several criteria. Among them, the bottom trawling survey carried out in 2006–2007, and non-parametric estimation methods such as Chao 2, Jackknife 1, Jackknife 2 and Bootstrap were used to estimate the fish species richness. The results of this case show that a total of 153 species of fish were caught in the trawling survey in the fourth quarter, belonging to 14 orders, 57 families and 101 genera. The 2006–2007 cruise is more complete for studying species richness. The Estimable expectations of fish species richness are: 250 (Chao 2), 204 (Jackknief 1), 241 (Jackknief 2) and 174 (Bootstrap). The number of fish species was significantly higher in summer and autumn than winter and spring. To manage fishery resources and sustainability in the sea area of Fujian Province, biological information and stock assessment are required. This meaningful information, especially for endemic and economically important species such as can set a baseline. Once species change exceeds the baseline range, it provides decision-making basis for marine biodiversity conservation and fisheries management.

**Keywords:** Minjiang Estuary; non-parametric estimation; richness; number of species

## 1. Introduction

Biologists generally define the sum of genes, species and ecosystems in an area as biodiversity [1]. There are four types of biodiversity including taxonomic diversity [2], ecological diversity [3], morphological diversity (genetic diversity and molecular diversity) [4] and functional diversity [5]. A wide range of aquatic organisms has been reported, including bacteria and unicellular algae to microscopic crustaceans, benthic animals, zooplankton and fish [6]. Species diversity and species richness have always been inseparable from the term "biodiversity" [7]. When studying a community, the first question is to know how many species the community consists of. Species diversity refers to the biodiversity at the species level, which is measured by the number and distribution characteristics of species in a certain space [8]. Species diversity mainly studies the status of species in a certain region from the perspectives of taxonomy, systematics and biogeography [9].

The loss of species diversity and resource decline is one of the important research topics [10]. The total number of species in a community, or species richness, is the oldest and most fundamental concept of diversity. The exact number of species can be used to estimate the rate of species extinction, which is crucial for biodiversity conservation [11]. Species richness is a direct measure of species diversity [12]. It is the basic unit of biodiversity research and the most commonly used measurement method in community ecology research [13]. Scientifically ascertaining the number of biological species is the key and

premise of biodiversity research. It has a decisive impact on the overall advancement of biodiversity research and the comprehensive reliability of research results [14]. At present, community species richness estimation mainly focuses on plant communities [15–18].

Fishes are the most abundant group of vertebrates. About 32,500 species have been discovered so far. It displays extensive genetic diversity and species richness. Fish are important components of aquatic ecosystems and are particularly vulnerable to severe disturbances from environmental changes [19]. At the same time, global warming directly affects the reproductive cycle of living organisms, accelerating the rate of species extinction [20]. Currently, the world is in the era of the sixth mass extinction [21].

In terms of fish species research, species richness mainly focuses on freshwater rivers and lakes [22–24]. Most efforts to estimate freshwater fish species richness emphasize the "stream reach" scale, typically a stream segment of several hundred meters or less, (for longer rivers, a sampling station with a length of several hundred meters is usually used to collect or evaluate the richness of freshwater fish) [25]. Unlike stream-reach scale surveys that encompass habitat diversity on spatial scales of pools, riffles, runs, and smaller, drainage basin or multiple drainage basin surveys must consider larger scale habitat heterogeneity. When determining species richness, it is necessary to consider the impact of environmental heterogeneity on fish distribution [25].

Less research has been done on fauna, especially marine life. Because independent biological surveys in the ocean are costly, they are easily restricted by on-site conditions. In reality, a complete survey of marine life is impossible. The typical method is to survey by sampling, that is, to survey a small part of an area to represent a larger area [26]. Accurate estimates of fish species richness are important as a basis for evaluating the effectiveness of conservation and management programs. Since the number of species is always underestimated by sampling, a variety of techniques have been used to estimate true species richness, including occupancy models and depletion methods [27–29]. The most commonly used technique is based on the species-area relationship (SAR), which describes the increase in the number of species that occurs as the sample area increases [29]. In addition, Mouillot et al. (2005) [30] adopted a new index proposed by Clarke & Warwick (2001) [31] based on the evenness of the distribution of taxa across the hierarchical taxonomic tree. Mouillot et al. assessed coastal lagoon quality with taxonomic diversity indices of fish, zoobenthos and macrophyte communities and identified the most affected regions [30].

Estuaries are the intersection zone of fresh water and seawater. Influenced by many factors such as river fresh water erosion, ocean tides, waves and currents, the physical and chemical conditions are complex, and the environmental factors are changeable. It is one of the most productive biomes in the world and an ecosystem with enormous resource potential and environmental benefits [32]. Also, estuarine is the area with the most frequent human activities, which provides many important goods and services for human society [33]. In recent years, due to chaotic management, unreasonable development and utilization, estuary siltation and transfer to land, etc., the ecological system has been seriously unbalanced and the ecological function has gradually declined [33].

To manage fisheries resources and sustainability in the sea area of Fujian Province, biological information and population assessments are needed. This study aims to analyze the fish species richness in the waters of the Minjiang Estuary. In this study, eight reports and surveyed data were collected and compared from 1990–2021 [34–41]. To obtain suitable analysis data, the content of the data was compared and judged. Therefore, this survey data during 2006–2007 were used to analyze the fish species richness in the Minjiang Estuary and adjacent waters. At the same time, some other survey and observation data in this sea area over the years were also analyzed to explore the relationship between the seasonal variation trend of fish abundance and related factors. The results hope to provide ideas for species richness, suggestions on how to improve conservation practices, and references for formulating management policies and fishery resources protection.

## 2. Materials and Methods

### 2.1. Research Design and Case Selection

To understand the fish species richness in the sea area of Fujian Province and propose management strategies in the future, several research conditions were considered. This includes important inshore fishing grounds, larger-scale surveys, appropriate analysis methods, etc. Therefore, the Minjiang River estuary was considered and selected as the study area. Some previous surveys and observations in this sea area are shown in Table 1.

**Table 1.** Number of fish species surveyed over the years in Minjiang Estuary and adjacent waters.

| Year | No. of Site | Spring (Per Season) | Summer (Per Season) | Autumn (Per Season) | Winter (Per Season) | Total | Reference |
|------|-------------|---------------------|---------------------|---------------------|---------------------|-------|-----------|
| 1990–1991 | / | 18 | 38 | 32 | 30 | 71 | ECCSIR [34] [1] |
| 2006 | 12 | 52 | 79 | 70 | 49 | 129 | Zhuang [35] |
| 2006–2007 | 12 | 67 | 97 | 84 | 64 | 153 | Wang et al. [36] |
| 2008 | 12 | 36 | 57 | / | / | 77 | Xu [37,38] |
| 2015 | 14 | 68 | 79 | 51 | 50 | 136 | Zhuang [35] |
| 2016 | 14 | 51 | 88 | 68 | 44 | 125 | Feng et al. [39] |
| 2017 | 11 | 36 | 51 | 41 | 31 | 111 | Zhang [40] |
| 2018 | 11 | 32 | 49 | 31 | 31 | 113 | Zhang [40] |
| 2021 | 14–16 | 58 | 91 | / | / | 104 | Liu et al. [41] |

Notes: [1] Editorial Committee of Comprehensive Survey of Island Resources in Fujian Province (ECCSIR).

The Minjiang River is the largest river in Fujian Province, with the geographic co-ordinates of 118°52′–119°58′ E and 25°17′–26°34′ N, covering an area of 6131 km$^2$. The Minjiang River estuary is a strong tidal delta estuary with mountain streams, and the annual average runoff to the sea is $620 \times 108$ m$^3$. In addition to seasonal runoff, the estuary area is also affected by the coastal waters of Fujian and Zhejiang, the Taiwan warm current, and the Kuroshio current [42]. The Minjiang Estuary is rich in fishery resources and diverse, and the species diversity is also representative in the Gulf of China. It is an important spawning ground for a variety of valuable economic species, as well as a traditional sea area for fixed net and drift gill net operations [43].

To obtain suitable analytical data, eight reports and survey data from 1990–2021 were collected and compared [35–41]. Interpretation of the suitability of fish species richness according to some of the above criteria. The main criteria to be assessed are the investment constraints of the survey, including survey time, cost effort, scope and completeness. We compared eight data documents (Table 1), and the number of survey sites in each cruise ranged from 11 to 16.

To compare the surveyed range and area of investigation, we extracted Zhang's literature [40] (Figure 1.2, p. 14). Different line labels in 1985: Fujian Provincial Fisheries Division Office [40] (green line); 2006–2007: Wang et al. [36] (purple line); 2008: Xu [37,38] (blue line); 2015–2016: Feng et al. [39] (red line); 2017–2018: Zhang's surveyed range and area [40] are the same as Feng et al. [39] (Figure 1). The sea area of surveyed in 2006–2007 is almost the largest.

### 2.2. Survey Site and Sampling Method

For a closer look at the cases selected for analysis, detailed sampling and surveyed sites settings are described below. A total of 12 sites had been set up in Minjiang Estuary and its adjacent sea areas (25°42′–26°27′ N, 119°42′–120°15′ E, about 3600 km$^2$) (Figure 1). The trawling surveys of fish species were carried out for 4 cruises respectively on 2–4 September (summer), 12–15 January (winter), 15–19 April (spring) and 19–24 October (autumn) during the period of 2006–2007.

During the survey, a single bottom trawler with a tonnage of 100 t and a main engine power of 221 kW was used. The fishing gear was a winged single bag trawl. Each net is towed for 1 h, and the towing speed is about 3–4 kn. The survey water depth is 10–50 m.

At the same time, the surface salinity and water temperature were measured, and wind speed and weather were recorded.

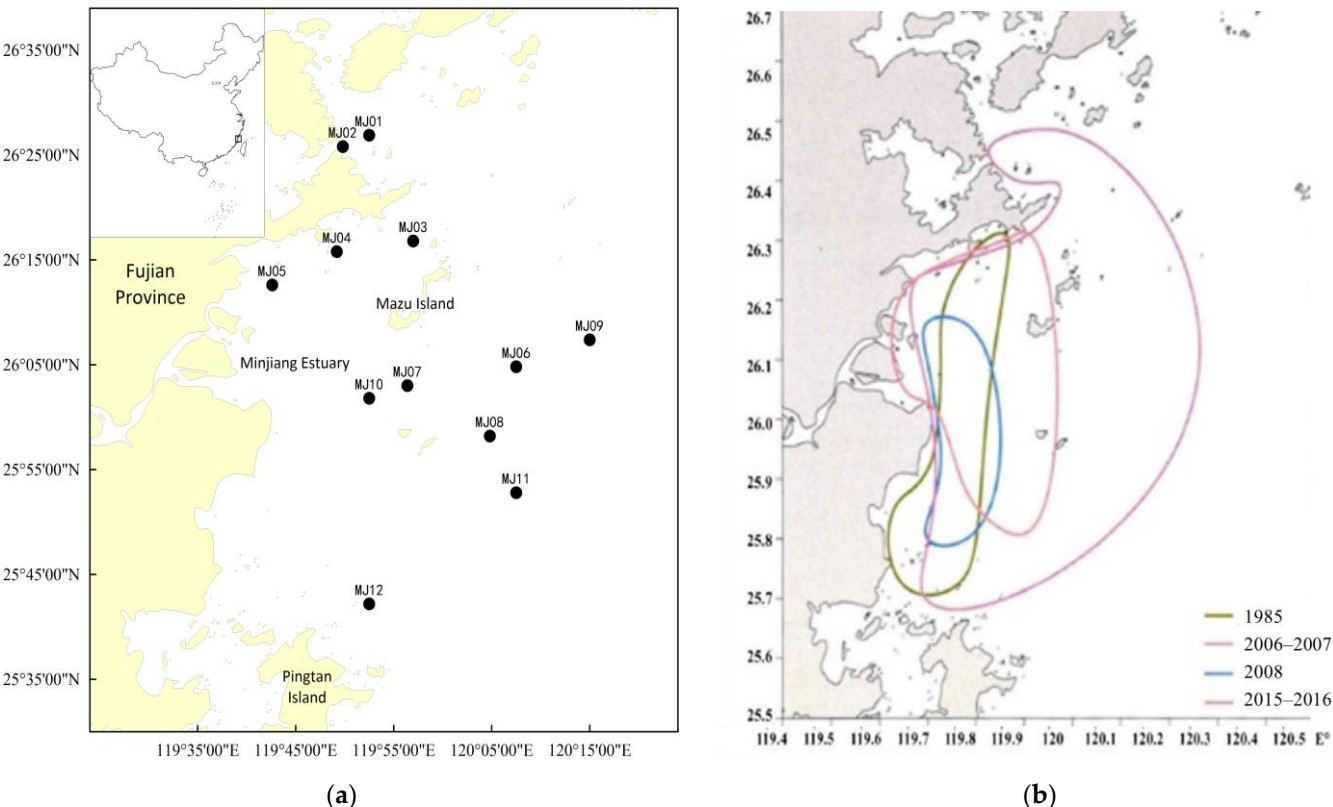

(**a**)                                                                (**b**)

**Figure 1.** Minjiang Estuary and its adjacent waters. (**a**) Bottom trawl sampling sites in this paper; (**b**) Historical fishery survey areas in the Minjiang Estuary and adjacent waters, Fujian Province. (Different line labels in 1985: Fujian Provincial Fisheries Division Office [40] (green line); 2006–2007: Wang et al. [36] (purple line); 2008: Xu [37,38] (blue line); 2015–2016: Feng et al. [39] (red line); Zhang's surveyed range and area [40] are the same as Feng et al. [39]. Figure excerpted from Zhang [40], p. 14).

### 2.3. Sample Collection and Processing

Sample processing refers to the "Marine Survey Specifications". After the net is lifted, the catch is poured on the deck, and the sundries are picked out. When the total weight of the catch is less than 30–40 kg, all samples are taken, and when the total weight of the catch is greater than 40 kg, after picking out rare and large specimens, randomly collect about 20 kg of samples. The samples were taken back to the laboratory for frozen storage, and the sub-sites identified, counted and weighed the samples.

### 2.4. Non-Parametric Estimation of Fish Species Richness

In order to understand the species richness in a certain region, some methods can be used to estimate or extrapolate it [42]. Usually, the non-parametric estimators of species richness mainly include Chao 1, Chao 2, ACE (an abundance-based coverage estimator), ICE (an incidence-based coverage estimator), Jackknife series and Bootstrap, and Michaelis-Menten Mean [43–50]. In this paper, the following non-parametric estimators are used to analyze the status of fish species richness in the Minjiang Estuary.

The Chao 2 estimator takes into account the number of rare species and uses the number of species that only appear in one or two samples:

$$\text{Chao}_2 = S_{obs} + \left( \frac{L^2}{2M} \right),$$ (1)

In Formula (1), $S_{obs}$ is the number of species appearing at the survey site. $L$ is the number of species that only appear in one site among all survey sites. $M$ is the number of species that only appeared in two sites among all survey stations [43].

The Jackknife estimator aims to reduce the bias of the estimate. It can be divided into first-order and second-order and multi-order knife-cut method estimators, and first-order and second-order estimators are often used [11]. The first-order jackknife 1 estimator is a function of the rare species in the recorded community, and the starting value is:

$$J_n^1(S) = S_{obs} + \left[ r_{1(1)}(n-1) \right] / n,$$
(2)

In Formula (2), $r_{1(1)}$ is the number of species that only appear at one site. $n$ is the number of investigation sites.

The second-order jackkinife 2 estimator is a function of the number of species that only appear in one site and the number of species that only appear in two sites, and its estimator is:

$$J_n^2(S) = S_{obs} + \left\{ \left[ r_{1(1)}^{(2n-3)/n} \right] - \left[ r_{1(2)}(n-2)^2 \right] / [n(n-1)] \right\},$$
(3)

$$r_{1(1)} = \sum_{i=1}^{n} r_{1i}$$
(4)

$$r_{1(2)} = \sum_{i<j}^{n-1} r_{1ij}$$
(5)

In Formula (3), $r_{1(1)}$ is the number of species that just appear in one site, and $r_{1(2)}$ is the number of species that happen in two sites. In Formula (4), $r_{1i}$ is the number of species that only appear in site $i$. In Formula (5), $r_{1ji}$ is the number of species that only appear in sites $i$ and $j$.

The formula for the Bootstrap estimator is:

$$B_n(S) = S_{obs} + \sum_{j=1}^{S_{obs}} (1 - Y_{\bullet j}/n)^n,$$
(6)

In Formula (6), $Y_{\bullet j}$ is the number of sites where type J appears.

The various estimates of fish species richness were calculated using Excel software 2020 and Biodiversity Pro 2.0 software [48].

## 3. Results

### 3.1. Composition of Fish Species Richness

From the four surveys in the Minjiang Estuary area, 153 species of fish were found, belonging to 2 classes, 14 orders, 57 families and 101 genera (Table 2). Perciformes had the largest number of families, genera and species, while Orectolobiformes, Torpediniformes and Gadiformes had only one species.

### 3.2. Changes in Fish Species Richness across Seasons

The number of fish species in Minjiang Estuary and nearby waters in each season is shown in Table 3. Of the 153 species of fish caught, 97 occurred in summer, 64 in winter, 67 in spring, and 84 in autumn. Accounted for 63.40%, 41.83%, 43.80% and 54.90% of the total, respectively. Among them, 52 species appeared in only one season in a year, accounting for 34.00% of the total number. The number of fish species is 14 appearing in two seasons, accounting for 9.15% of the total. There are 26 species of fish in the four seasons, accounting for 17.00%. Most of them are gobies and flatfishes with poor swimming ability.

**Table 2.** Fish species composition in the Minjiang Estuary and its adjacent waters.

| Class | Order | Family | Genus | Species |
|---|---|---|---|---|
| Chondrichthyes | Orectolobiformes | 1 | 1 | 1 |
| | Rajiformes | 3 | 3 | 4 |
| | Myliobatiformes | 2 | 2 | 6 |
| | Torpediniformes | 1 | 1 | 1 |
| Actinopterygii | Clupeiformes | 2 | 8 | 16 |
| | Myctophiformes | 2 | 3 | 4 |
| | Anguilliformes | 5 | 6 | 6 |
| | Siluriformes | 2 | 2 | 2 |
| | Gadiformes | 1 | 1 | 1 |
| | Mugiliformes | 2 | 4 | 7 |
| | Perciformes | 29 | 61 | 81 |
| | Pleuronectiformes | 3 | 4 | 14 |
| | Tetraodontiformes | 2 | 3 | 7 |
| | Lophiiformes | 2 | 2 | 3 |
| Total | | 57 | 101 | 153 |

**Table 3.** Number of fish species surveyed at different seasons in Minjiang Estuary and adjacent waters.

| Cruise | Season | Surveyed Period | Site No. | Species No. | Species No. [1] | Species No. [2] |
|---|---|---|---|---|---|---|
| 1 | Summer | 2 September 2006–4 September 2006 | 12 | 97 | 23 | 6 |
| 2 | Winter | 12 January 2007–15 January 2007 | 12 | 64 | 12 | 7 |
| 3 | Spring | 15 April 2007–19 April 2007 | 12 | 67 | 4 | 6 |
| 4 | Autumn | 19 October 2007–24 October 2007 | 12 | 84 | 13 | 5 |
| Total | | | 48 | 153 | 52 | 14 |

Note: [1] Number of species that occur only once. [2] Number of species that occur only twice.

### 3.3. Non-Parameter Estimation of Fish Species Richness

Fish richness estimated for the four seasons using four non-parametric methods is shown in Figure 2. Overall, the average fish richness of the four seasons calculated by the Chao 2 method is the highest, with 136 species. It was followed by Jackknief 2 and Jackknief 1 with 127 and 108 species, respectively. The fewest is 91 as estimated by the Bootstrap method. The results estimated by these four methods are higher than the actual observed values. Among them, Chao 2 method estimated the largest number of species in autumn, with 161 species. The number of fish species estimated by the Bootstrap method is the least in winter, with 75 species. Among the four seasons, the fish richness estimated by Jackknief 2, Jackknief 1 and Bootstrap methods is the highest in summer. However, the fish richness estimated by the Chao 2 method is relatively the highest in autumn. This is because the ratio of the number of fish species with two sites in autumn to the number of fish species with one site is the smallest, which is 0.229. The values for spring, summer and winter were 0.259, 0.359 and 0.286, respectively.

The number of fishes present at the 12 survey sites are shown in Table 4. Among them, site MJ 10 has the largest number of fish species, 68 species, and site MJ 9 has the least number of fish species, 48 species. There is a total of 5 sites with more than 60 species, namely site MJ 3, MJ 4, MJ 6, MJ 10 and MJ 12. The high number of fish species at these sites may be related to the surrounding environment. These 5 sites show that they are all in the outer waters of the Minjiang Estuary, not far from the island, and have relatively many rocks and reefs.

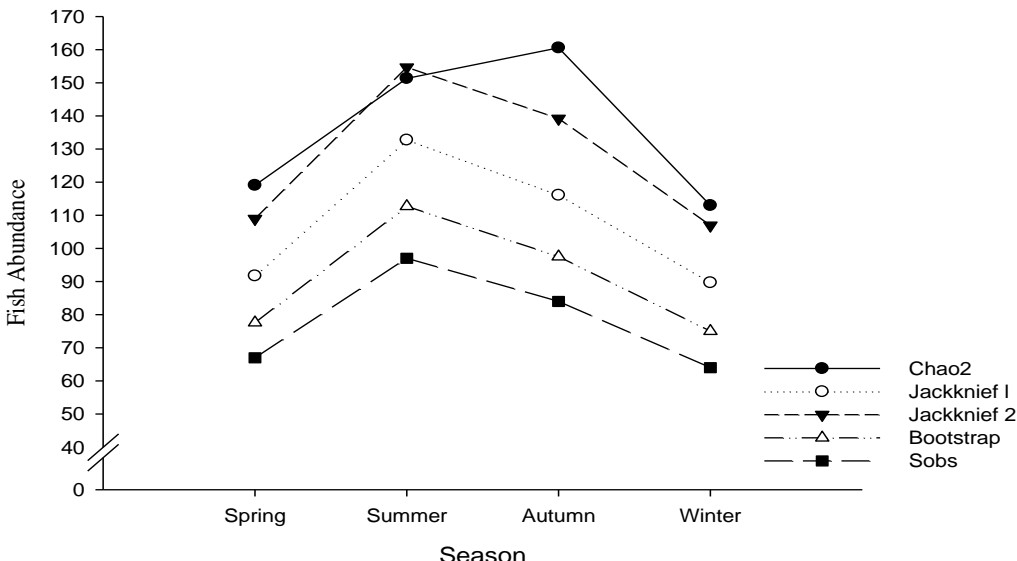

**Figure 2.** Comparison of four seasons' fish species richness estimated by non-parametric estimation model in the Minjiang Estuary and its adjacent waters.

**Table 4.** Number of fish species inventoried at different sampling sites in Minjiang Estuary and adjacent waters.

| Site | Spring | Summer | Autumn | Winter | Total |
|------|--------|--------|--------|--------|-------|
| MJ1 | 17 | 30 | 21 | 25 | 54 |
| MJ2 | 13 | 32 | 28 | 8 | 49 |
| MJ3 | 17 | 41 | 34 | 23 | 63 |
| MJ4 | 30 | 20 | 25 | 17 | 65 |
| MJ5 | 30 | 20 | 17 | 15 | 51 |
| MJ6 | 20 | 22 | 22 | 24 | 60 |
| MJ7 | 16 | 26 | 28 | 16 | 54 |
| MJ8 | 14 | 27 | 25 | 21 | 54 |
| MJ9 | 18 | 17 | 25 | 29 | 48 |
| MJ10 | 15 | 33 | 32 | 15 | 68 |
| MJ11 | 18 | 22 | 27 | 22 | 54 |
| MJ12 | 16 | 29 | 24 | 25 | 62 |
| Total | 67 | 97 | 84 | 64 | 153 |

In summer, *Johnius belengerii*, *Polydactylus sextarius* and *Harpadon nehereus* were the dominant species, and the occurrence rate of *J. belengerii* and *P. sextarius* were up to 100%. In addition, more than 20 fish species including *Lagocephalus spadiceus*, *Apogon lineatus*, *Apogon quadrifasciatus* and *Lepturacanthus savala* had a higher occurrence rate than 40%, which were common species in the Minjiang estuary in summer.

In autumn, *P. sextarius* and *H. nehereus* were the dominant species, with 100% occurrence rate. In addition, more than 17 fish species including *L. spadiceus*, *Pampus argenteus* and *Eleutheronema rhadinum* had a higher occurrence rate than 40%, which were common species in the Minjiang estuary in autumn.

In winter, *Collichthys lucidus*, *Coilia mystus*, *H. nehereus* were the dominant species, and the occurrence rate of *C. lucidus* and *C. mystus* were up to 100%. In addition, more than 15 fish species including *Arius sinensis*, *P. argenteus*, *Pseudosciaena polyactis* and *Chaeturichthys stigmatias* had a higher occurrence rate than 40%, which were common species in the Minjiang estuary in winter.

In spring, the *C. lucidus* and *Parargyrops edita* were the dominant species. In addition, more than 9 fish species including *Trachurus japonicas*, *Thryssa vitrirostris* and *Thryssa*

*kammalensis* had a higher occurrence rate than 40%, which were common species in the Minjiang estuary in spring.

The similarity of fish composition in the four seasons is low, indicating that the seasonal variation is large.

The fish species richness was higher in summer and autumn than in winter and spring (Figure 2). This difference is related to the migratory characteristics of some fishes, such as: *Siganus fuscescens*, *Gastrophysus spadiceus*, and *Fugu xanthopterus*, etc. They will only appear in the Minjiang Estuary and its surrounding areas in summer and autumn. The seawater temperature is higher in summer and autumn, which is the spawning season for most migratory fish. At this time, a large number of various migratory fish migrate from the deep sea to the estuary of the shallow sea to spawn. As a result, the fish species richness in the Minjiang Estuary in summer and autumn is much greater than that in winter and spring. It can be seen that there are differences in fish richness in different seasons in the Minjiang Estuary.

The relationship between fish species richness and sea surface temperature is shown in Figure 3. The average sea surface temperature in this sea area varies greatly throughout the year. The sea surface temperature is the lowest in winter with an average of 12.9 °C, and the highest in summer with an average of 29.8 °C. The richness of fish species varies with the sea surface temperature in each season, showing a very obvious positive correlation ($R^2 = 0.9632$). That is, the water temperature is higher in summer and autumn, and the richness of fish is also higher. In spring and winter, the water temperature is lower, and the fish species richness is also lower.

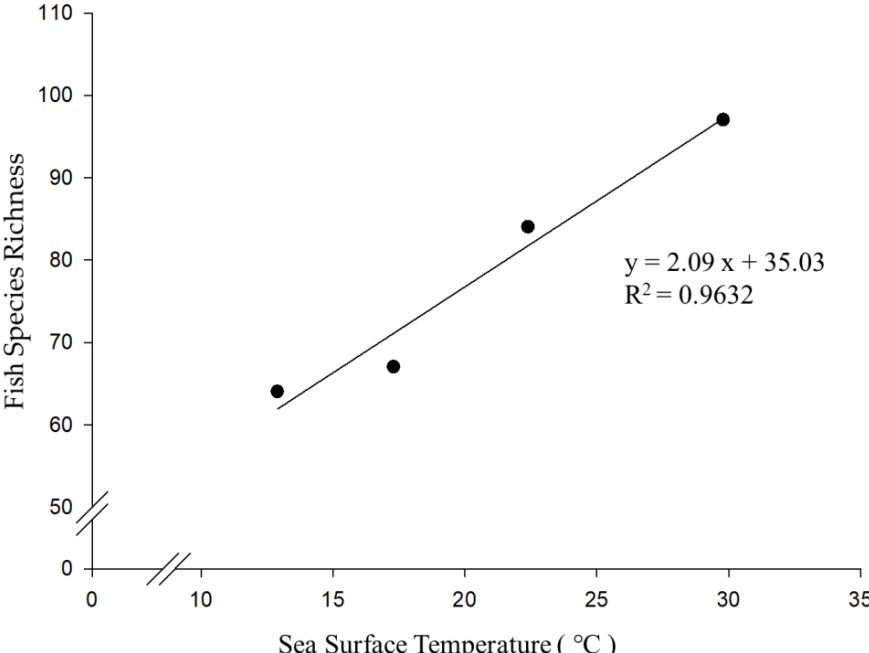

**Figure 3.** The relationship between fish species richness and sea surface temperature (SST) in the Minjiang Estuary and its adjacent waters. The black circles are the coordinates of the average surface temperature and the species number within seasons; the line is the best fit line for a linear regression model.

Annual fish species richness was estimated by four non-parametric methods and $S_{obs}$: 250 (Chao 2), 204 (Jackknief 1), 241 (Jackknief 2), 174 (Bootstrap) and 153 ($S_{obs}$) (Figure 4). These estimates were all greater than the actual number of species which was 153. The species numbers estimated by Chao 2 and Jackknief 2 were close, but higher than those estimated by the other two methods. The estimated value of Bootsrtap method is the lowest, and it is also the closest to the actual number of fish species. According to

the average value of these four methods, the fish richness of the Minjiang Estuary and its adjacent waters can be estimated to be 217 species.

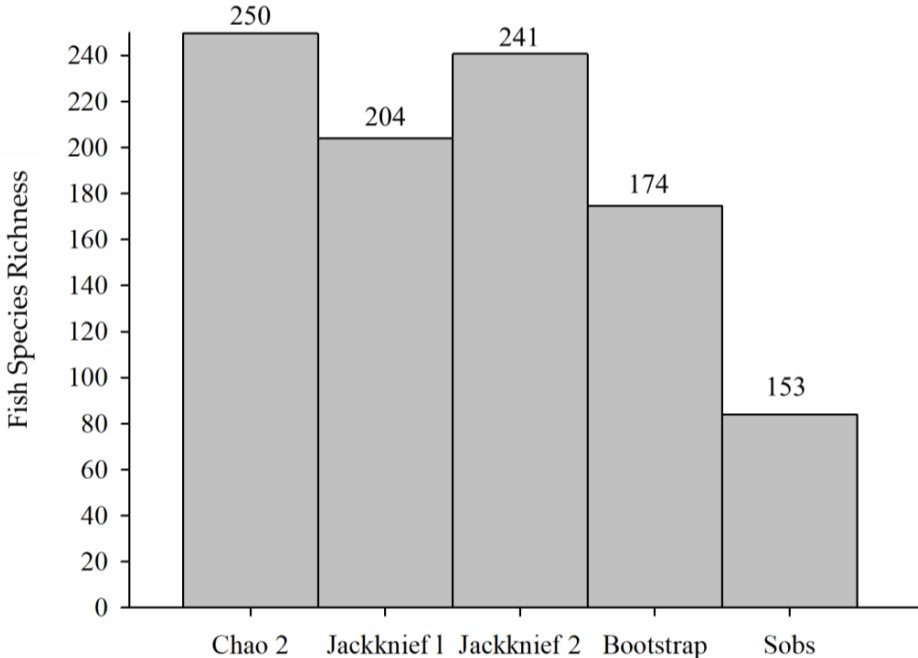

**Figure 4.** The fish species richness estimated by four non-parametric estimation model and $S_{obs}$ in the Minjiang Estuary and its adjacent waters.

*3.4. The Relationship between Fish Richness and Sampling Location*

In Figure 5, there is a power exponential function relationship between the fish species richness and the sampling sites number ($R^2 = 0.9820$). Curves were fitted using data from all 12 sites (solid dots), and then fish species richness was estimated from 13 to 24 site number (open dots). Finally, according to the four methods, the fish species richness is estimated to be 217 species with 24 measuring sites.

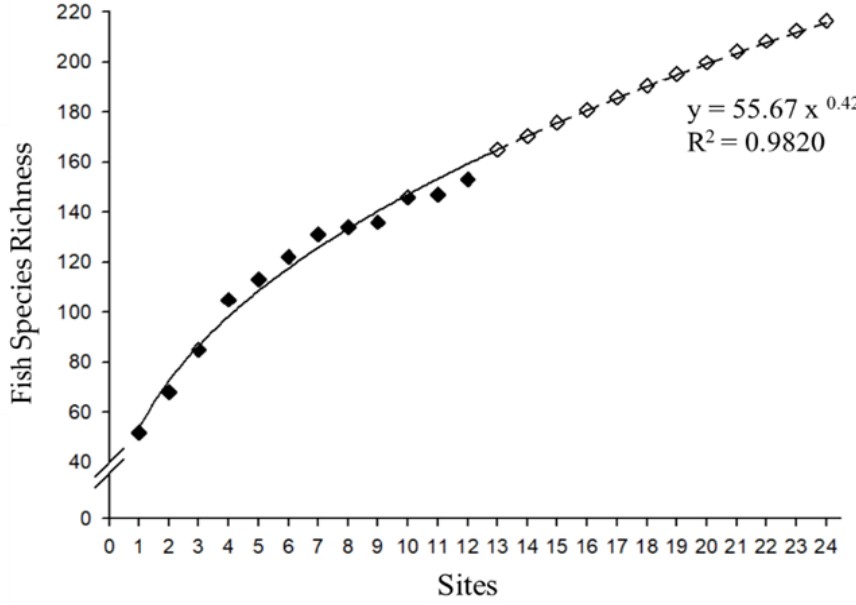

**Figure 5.** The accumulation curve of fish species richness with the number of stations in the Minjiang Estuary. The black diamonds are the real species richness in each site; the white diamonds are the predicted species richness when added to 24 sampling site.

## 4. Discussion

### 4.1. The Role of Non-Parametric Estimates of Fish Species Richness

Traditional curve fitting methods include negative exponential function, Weibull function, logistic function, and Michaelis-Menten function, using parametric curves to fit species accumulation or species area curves to predict their asymptotes [51,52]. Not based on any statistical sampling model is a problem with curve fitting methods. So, the variance of the resulting asymptotes cannot be assessed without assumptions. Furthermore, rigorous statistical comparisons of estimators between fish community cannot be performed [52]. The non-parametric method does not make any assumptions about the mathematical form of the underlying species abundance distribution, avoids the above disadvantages, and is more robust in application [52].

Non-parametric estimation methods provide a good way to study the problem of species richness. Because the non-parametric estimation method does not require any assumptions about the distribution of species, it just uses computer algorithms instead of traditional mathematical methods to obtain the solution of the number of species [49]. In the case of a small number of sample squares, it shows its superiority over traditional statistical methods. They only need to consider the presence of species, not abundance data. At the same time, in the sampling survey, the mutual influence of species within the sample plot is not considered, and generally better results can be obtained [46]. Most nonparametric estimates of the number of undetected species are based on counts of detected rare species, especially singletons and doublets for abundance data, or unique and duplicates for occurrence data [52]. The disadvantage is that the number of species estimated will not exceed twice the number of species sampled. Therefore, good results can be obtained with these methods only if the number of sampled species is quite large, or if the number of sparse species is not too large compared to the total number of species [49].

In the survey of the number of fish species in the Minjiang Estuary, sparse species accounted for 34% of the total, accounting for nearly one-third of the total. Moreover, the number of species sampled is not large enough, so the fish species richness estimated by the non-parametric estimation method is equal to the real fish species richness (the number of fish species recorded in the history of the Minjiang Estuary and its adjacent waters is equal to 372 species [53]).

### 4.2. Data Representativeness and Implications for Fish Species Richness Estimates

Data representativeness and its impact on fish species richness estimates are critical. Among which, the Fujian Provincial Government conducted a large-scale survey during 2006–2007 in the Minjiang Estuary [36]. In terms of estimating fish species rich-ness, this data demonstrates the completeness and adequacy of the survey. According to Table 1, it can also be found that the total number of fish species in 2006–2007 is also the largest, except for the 67 species in the spring, which is lower than the 68 species in the spring of 2015–2016 [39]. The number of fish species found in other seasons is higher than that in other years. The number of fish species surveyed is higher. Apparently, the survey data from 2006 to 2007 is the most comprehensive survey of fishery resources in the Minjiang Estuary, which is particularly important and representative.

In stark contrast to adjacent marine and freshwater ecosystems, knowledge of patterns and predictors of fish species richness in estuaries is poor. This is because surveys in estuaries are fragmented and often based on area-specific approaches, without standardization of sampling effects and mismatched spatial extents [54–56]. In general, the factors affecting fish species richness at the spatial scale are divided into two parts. Globally, fish species richness varies across marine biogeographic domains and increases with mean sea surface temperature, terrestrial net primary productivity, and stability of connectivity to open versus temporarily open estuaries. At the regional scale, fish species richness increases with estuary area and continental shelf width [57].

Liu and Ma [58] pointed out that the most important factor affecting the fish species richness is the number of survey sites. On a larger spatial scale, the fish species richness

increases with the increase of the surveyed water depth and area, which can generally be expressed as a species richness-area relationship. Pasqua et al. [59] mentioned that the main factors affecting the fish species richness of estuaries are latitude, area, width, runoff and intertidal zone size. Marin et al. [60] also described that one of the most important factors for predicting species richness is time-integrated area. Apparently, diversity and diversification rates can be estimated over large scales by looking at species time-integrated areas of area and energy over time [61]. The hypothesis of species–area relationships (SAR) assumes a change in species numbers with increasing area and has been termed one of the few laws in ecology [57].

The study of fish species richness in the ocean cannot set a sampling site at intervals of several hundred meters like in rivers. It can only be used to set several sampling sites in the research area according to the water depth or terrain characteristics, or to make predictions on a larger scale, such as an ocean or a global scale [62]. Estuaries, as the confluence of fresh water and sea water, are affected by factors such as river fresh water and ocean tides, waves, and ocean currents. The physical and chemical conditions are complex and the environmental factors are changeable. It exhibits large variations in water temperature, salinity, turbidity, and habitat types. The changeable characteristics provide the necessary ecological conditions and living space for the growth and reproduction of many fishes, and it is one of the highest-producing biological communities in the world [63]. Moreover, human activities in the estuary area are the most intensive, and the chaotic management and unreasonable development lead to drastic changes in the estuary environment and imbalance of the ecological system. Coupled with overfishing, the fish in the estuary are facing serious threats, and many species have disappeared [64,65].

Like estuaries, the abiotic and biotic variables of coastal lagoons are highly heterogeneous in space and time, and this heterogeneity complicates the assessment of their ecological status. It is critical to monitor and protect these fragile ecotones and the resources and services they sustain [66]. Bremer et al. [66] emphasized the need for a multifaceted framework to properly assess lagoon conditions and highlighted the need for high-frequency lagoon monitoring to avoid erroneous condition assessments and resulting management plans.

According to Table 1, it can be found that the data of 2006–2007 including the number of stations, seasonal conditions and species numbers were complete and suitable [40]. Since 2006–2007, we also found that in each survey, the total number of fish species collected showed a downward trend when the survey sites were not much different. Such trends may be related to anthropogenic fishing, environmental factors and the number of surveys. Therefore, the survey data from 2006 to 2007 is the most comprehensive survey of fishery resources in the Minjiang Estuary since the beginning of the 21st century, which is particularly important and representative.

The 153 species of demersal fish collected this time were higher than the research results of two quarters in 2008 and 2009 [37,38]. It is also higher than the results of four different surveys in the sea area from 2015 to 2019 [35,39–41]. The number of fish species that appeared in the sampling survey showed a relatively large gap with the number of fish species that appeared in previous studies. A total of 372 fish species have been recorded in the Minjiang Estuary and its adjacent waters [53]. However, only 153 species were found in this sampling survey.

Regarding the estimation of Chao 2 estimator (Formula (1)), Jackknife estimator (Formula (2)), second-order jackkinife 2 estimator (Formula (3)) and Bootstrap estimator (Formula (6)), it is mainly based on the $S_{obs}$ + adjust item. That is, the influence of fish species richness estimation is mainly on the main item $S_{obs}$. We observed $S_{obs}$ from eight data documents ($S_{obs}$ = "Total" item in Table 1), which also shows that the more "Total" value, the higher fish species richness value. In this study, the maximum value of $S_{obs}$ is 153 and the maximum estimate of Chao 2 estimator is 250. So that, the maximum value estimated by non-parametric estimation method is 250. Although the maximum value has been obtained, it is less than half of the historical record.

The possible reasons are as follows: First, due to global warming, human activities, environmental pollution [67,68] and other factors, certain fish species are not suitable for habitat in this area or even become extinct. Second, the systematic error caused by sampling survey, so the systematic error caused by random factors seems to be unavoidable in sampling statistics. Third, it may be due to the limitation of the survey station or area that some fish cannot be collected. The survey used bottom trawling to collect samples. The fish collected basically lived in the bottom layer, while the middle- and upper-layer fish were not easy to collect. In addition, due to the selectivity of the survey nets, many fish cannot be caught with a single net. Some fast-swimming fish, small fish and rare species are difficult to catch.

To sum up, estimates of fish species richness in estuary cannot be as high as in rivers. The environmental conditions and influencing factors of the estuary are more complex and changeable. Therefore, corresponding survey methods should be adopted according to the living habits of different fishes, and the sampling survey process should be improved. For example, arrange more sites, change quarterly sampling to monthly sampling, and expand the sampling area. At the same time, the site layout should consider different environmental factors such as water depth, water temperature, seabed topography, and latitude.

### 4.3. Formulate a Strategy for the Protection of Regional Species

The results of this study come from data that goes back almost 20 years. This is important for developing strategies to protect local biodiversity. Apparently, the results of this study reveal the possibility of fish species disappearing. The threat of human activities to biological species resources is becoming more and more serious, and it is imperative to formulate corresponding regional biological species resources protection strategies [67]. The prerequisite condition is to clearly grasp and understand the change of species richness in the research sea area. Therefore, follow-up sampling surveys must be carried out regularly, and the sampling survey area should be expanded as much as possible if conditions permit. It should also consider the impact of land use changes, environmental pollution, major projects, etc. on marine biological species. Especially the impact on endemic species and economically important species provides decision-making basis for marine biodiversity conservation planning. It is even possible to propose new sampling techniques, such as complementary ones, which would allow catch data to be done in a more quantitative manner and consider combining catch data with fixed sonar monitoring [69].

Governments assess the likely contribution of these approaches to conservation management goals before devising improvements and need to be judged as having good potential to implement plans effectively and responsibly [70]. It is widely recognized that there is a need to reduce fishing effort and restore habitat to increase the resilience of biological conservation [63,64]. At this juncture, the scientific basis for marine restocking, stock replenishment, and sea ranching continues to advance at a rapid pace. This is a good time to implement relevant conservation policies for endangered species. Therefore, Wang et al. [71] have put forward suggestions for appropriately reducing fishing effort, including severely cracking down on illegal fishing activities, carrying out legal and ecological education, increasing inspections, and organizing professionals from relevant scientific research institutions to regularly carry out publicity and education on marine biological resources and environmental protection. The above four directions are quite practical and meaningful for improving the ecological awareness of protecting and conserving estuaries.

### 5. Conclusions

The estuary is affected by many factors such as river freshwater erosion, ocean tides, waves, currents, etc. The physical and chemical conditions are complex and the environmental factors are changeable. Scientifically ascertaining the number of biological species is the key and prerequisite for biodiversity research. The comprehensive reliability of species richness estimation studies in the Minjiang Estuary has a decisive impact on improving

conservation practices, formulating management policies, and protecting fishery resources. A total of 153 species of fish were identified in the four-quarter trawling survey, belonging to 2 classes, 14 orders, 57 families and 101 genera. According to the number of fish species that appeared in the four cruise surveys, the estimated fish richness was: 250 (Chao 2). There were differences in fish species richness in different seasons. The number of fish species in summer and autumn was significantly higher than that in winter and spring. The results of this study reveal the possibility of fish species disappearing and not appearing. The environmental conditions and influencing factors of estuaries were more complex and changeable. In the future, corresponding investigation methods should be adopted to improve according to the living habits of different fishes. It can be seen from the eight reports that the environmental conditions and influencing factors of estuaries are more complex and changeable. In the future, according to the living habits of different fishes, corresponding investigation methods should be adopted for improvement.

**Author Contributions:** Conceptualization, J.-Q.W., J.L., Y.-J.S., T.-J.C. and L.-M.H.; methodology, J.-Q.W. and Y.-J.S.; software, J.-Q.W., J.L., Y.-J.S. and L.-M.H.; validation, J.-Q.W., J.L., Y.-J.S. and L.-M.H.; formal analysis, J.-Q.W. and Y.-J.S., investigation, J.-Q.W., J.L., Y.-J.S., L.-M.H. and X.-R.W.; resources, J.-Q.W., Y.-J.S., L.-M.H. and X.-R.W.; data curation, J.-Q.W., Y.-J.S., L.-M.H. and X.-R.W.; writing—original draft preparation, J.-Q.W., T.-J.C. and Y.-J.S.; writing—review and editing, J.-Q.W. and Y.-J.S.; visualization, J.-Q.W., J.L., Y.-J.S., L.-M.H., T.-J.C. and X.-R.W.; supervision, J.-Q.W. and Y.-J.S.; project administration, J.-Q.W.; funding acquisition, J.-Q.W., J.L and Y.-J.S. All authors have read and agreed to the published version of the manuscript.

**Funding:** This work was supported by the Fujian Provincial Department of Education grant number JAT190347 and JT180272; Fujian Provincial Department of Science and Technology grant number 2021J01825; National Science Foundation of Fujian Province (Grant No. 2020J05136). The funders had no role in study design, data collection and analysis, decision to publish, or preparation of the manuscript.

**Data Availability Statement:** Not applicable.

**Acknowledgments:** This project was supported by the Fujian Provincial Department of Education (JAT190347 and JT180272), Fujian Provincial Department of Science and Technology (2021J01825) and the Cultivation program of National Natural Science Foundation of Jimei University (ZP2020023). We would like to thank Y.C. Kuo for his contributions to the comments on the manuscript. We would also like to thank the anonymous reviewers, whose useful suggestions were incorporated into the manuscript.

**Conflicts of Interest:** The authors declare no conflict of interest.

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
