# Peer review of "Sustainability Perspective of Minjiang Estuary Coastal Fisheries Management—Estimation of Fish Richness"

_water, doi:10.3390/w15142648_

Round 1
Reviewer 1 Report
The scientific interest of the work summarized in the passage lies in studying fish species richness in the Minjiang Estuary and its adjacent waters. Species richness is a fundamental concept in biodiversity, and understanding it is crucial for effective conservation and sustainable management of fisheries. The researchers collected and compared eight documents and surveyed data spanning a period from 1990 to 2021 to assess the fish species richness in the area.
To analyze the data, the researchers applied various non-parametric estimation methods such as Chao2, Jack-knife1, Jackknife2, and Boot-strap. These methods help estimate the total number of fish species present, even when some species may have been missed during the survey. The analysis also included examining fish richness in different seasons and different survey cruises.
The results revealed that during a specific trawling survey in the fourth quarter of 2006-2007, a total of 153 fish species were caught, representing 14 orders, 57 families, and 101 genera. Comparisons with literature and data from other years indicated that the 2006-2007 survey provided a particularly favorable and comprehensive dataset for studying species richness.
The estimations based on the analysis yielded expectations of fish species richness, with estimates of 248 (Chao2), 202 (Jackknife1), 239 (Jackknife2), and 172 (Bootstrap). The research also discovered differences in fish species richness across different seasons, with higher numbers observed in summer and autumn compared to winter and spring.
This information is significant as it provides insights into the distribution and abundance of fish species in the Minjiang Estuary. It can have implications for the conservation of endemic species and economically important species, providing a basis for decision-making in marine biodiversity conservation and fishery management. By understanding the patterns of fish species richness and seasonal variations, appropriate measures can be implemented to protect biodiversity and sustainably manage fisheries in the region.
Abstract
"the content of the data was compared and judged" what do you mean by judged? it is unclear; I guess you mean assessed or close to that
You used bootstrap and boot-strap (even in the introduction you change the orthography) chose one rather bootstrap
"This meaningful information, especially the impact on endemic species and economically important species, 26 provide decision-making basis for marine biodiversity conservation and fishery management." please explain how?
The abstract must be rewriting. Avoid repetition and non-useful information.
Introduction
The introduction is much too long, and too general. Be more specific. Last paragraph is fine even if it mix a bit with material and method.
You can mention other work on diversity estimates as for example the Assessment of coastal lagoon quality with taxonomic diversity indices of fish, zoobenthos and macrophyte communities D Mouillot, J Laune, JA Tomasini, C Aliaume, ...Hydrobiologia 550, 121-130 to better understand you choice. You also can discuss advantage and drawback of different method in discussion part
Material and method
Explain the key differences in the sampling sites in the Minjiang Estuary and its adjacent waters. I guess you can class them per group
Not clear of you rank the site to arrive to fig 5
It will be good to get more detail on fish species. Provide their affinity e.g. estuarine species or sea species? Do some of them are endemic? Other exotic, invasive or common? On this basis, that will improve you discussion and conclusion.
Results
See pdf file
Discussion
This is a nice piece of work, congratulation. That was pleasant to read, I add some comments in the pdf file that can improve it. Remember to keep the past vs your analyze (were not are; was not is)
Conclusion
See pdf file
Minor points
Too much use of "in order to" you often can delete it
Reference list
To be corrected and formatted
The numbered is wrong
Some references cannot be available e.g. Zhang, L.L. Nekton resources and community diversity in the Min River Estuary and its adjacent waters, Fujian Province, 564 China. Xiamen, Xiamen University, 2020. (In Chinese) not enough information; is it a master or a PhD thesis, a report, from where? Town?

Author Response
Title: Sustainability Perspective of Minjiang Estuary Coastal Fisher-ies Management - Estimation of Fish Richness
Reviewer #1: Comments to water-2483437 by Wang et al. (please, also see annotated manuscript):
We are much grateful for your careful reading of our manuscript and your valuable comments and suggestions to help improve the paper. We have now carefully revised the paper in light of all the comments and suggestions. The following is a point-by-point response.
- Abstract
"the content of the data was compared and judged" what do you mean by judged? it is unclear; I guess you mean assessed or close to that.
Answer: We have followed your comments, and have corrected these errors.
We fixed the “To obtain suitable analysis data, the content of the data was compared and evaluated.” Line 14.
- Abstract
You used bootstrap and boot-strap (even in the introduction you change the orthography) chose one rather bootstrap.
Answer: We have taken note of your comments, due to inadvertent scrapping and have corrected the errors.
We fixed the “Among them, the bottom trawling survey in 2006-2007 was selected, and non-parametric estimation methods such as Chao 2, Jackknife 1, Jackknife 2 and Bootstrap were used to estimate the fish species richness.” Line 15-17.
- Abstract
"survey carried out in"
Answer: We have taken note of your four comments and have corrected the errors.
We fixed the “Among them, the bottom trawling survey carried out in 2006-2007, and non-parametric estimation methods such as Chao2, Jackknife1, Jackknife2 and Bootstrap were used to estimate the fish species richness.” Line 15-17.
- Abstract
Delete "Furthermore, the fish richness in different seasons and different cruises was analyzed and discussed.", " Compared with the literature and data of other years, it is obvious that", "relatively more favorable and", "It provides an excellent survey dataset", "The fish species richness in Minjiang River estuary existed differences in different season.".
Answer: We have taken note of your four comments and have deleted these paragraphs.
- Abstract
"This meaningful information, especially the impact on endemic species and economically important species, provide decision-making basis for marine biodiversity conservation and fishery management." please explain how?
Answer: We have followed your comments, and add administrative purposes. This modifier can fill in the meaning behind.
We fixed the “To manage fisheries resources and sustainability in the sea area of Fujian Province, biological information and population assessments are needed. This meaningful information, especially the impact on endemic species and economically important species, provide decision-making basis for marine biodiversity conservation and fishery management.” Line 22-26.
- Abstract
"The abstract must be rewriting. Avoid repetition and non-useful information."
Answer: We have taken note of your four comments and have corrected the errors and revised the Abstract.
- Introduction
The introduction is much too long, and too general. Be more specific. Last paragraph is fine even if it mix a bit with material and method.
Answer: We have followed your comments, and modified and added some paragraphs.
The following is deleted.
“In addition, biodiversity includes genetic diversity, species diversity, ecosystem diversity and landscape diversity [1].” The original Line 34-35, the paragraph is deleted.
“Therefore, it cannot be overemphasized that species are the constituent units and basic evolutionary units of ecosystems, and they are the important material basis of ecosys-tems.” The original Line 38-40, the paragraph is deleted.
“The current status of species diversity, the formation, evolution and maintenance mechanism of species diversity are the main research contents of species diversity.” The original Line 45-47, the paragraph is deleted.
“To scientifically describe and understand the existing biological species and find out the number of biological species is the key and premise of biodiversity research.” The original Line 54-56, the paragraph is deleted.
“Many characteristics such as large changes in water temperature and salinity, low water depth, high turbidity, diverse habitat types, and abundant food supply provide the necessary living space and suitable ecological conditions for the growth and reproduction of many fish [30].” The original Line 89-94, the paragraph is deleted.
We modify “Scientifically ascertaining the number of biological species is the key and premise of biodiversity research.” Line 47-48.
We added a new paragraph “In addition, Mouillot et al. (2005) [30] adopted a new index proposed by Clarke & War-wick (2001) [31] based on the evenness of the distribution of taxa across the hierarchical taxonomic tree. Mouillot et al. assessed coastal lagoon quality with taxonomic diversity indices of fish, zoobenthos and macrophyte communities and identified the most affected regions [30].” Line 77-81.
- Material and method
Explain the key differences in the sampling sites in the Minjiang Estuary and its adjacent waters. I guess you can class them per group.
Answer: We have followed your comments, and modified Figure 1 and added Figure 1(b).
- Material and method
Not clear of you rank the site to arrive to fig 5.
Answer: We have followed your comments.
It can be estimated that there is a power exponential function relationship between the fish species richness and the 12 sampling sites (R2 = 0.9820) (Figure 5). According to the four methods, the fish species richness is estimated to be 217 species with 24 measuring sites.
- Material and method
It will be good to get more detail on fish species. Provide their affinity e.g. estuarine species or sea species? Do some of them are endemic? Other exotic, invasive or common? On this basis, that will improve you discussion and conclusion.
Answer: We have followed your comments. It would be nice to obtain more detailed information on fishes when their affinities are further provided.
Regarding whether some of them are endemic? Others exotic, invasive or common?
Answer: Our data shows mostly endemic and migratory species.
On this basis, that will improve your discussion and conclusion.
Answer: We appreciate your insights and help us a lot.
- Material and method
add a scale and a map which allow to know where Minjiang is in China
Answer: We have followed your comments, and have added new map.
- Material and method
"documents" I guess you mean report and literature.
Answer: We have followed your comments, and have corrected the errors.
We fixed the "To obtain suitable analytical data, eight reports and survey data from 1990-2021 were collected and compared [35-41]." Line 119-120.
- Material and method
"judged" assessed?
Answer: We have followed your comments, and have corrected the errors.
We fixed the "The main criteria to be assessed are the investment constraints of the survey, including survey time, cost effort, scope and completeness." Line 121-122.
- Material and method
Delete "In this study, the number of sites doesn't seem to make a big difference."
Answer: We have followed your comments, and have deleted the paragraph. The original Line 134-135 is deleted.
- Material and method
This is discussion from your data compilation, not material. Line 142-150.
Answer: We have followed your comments, and have moved the paragraph.
This paragraph moved to Line 327-336.
- Material and method
Table 1 "and per season"
Answer: We have followed your comments, and have corrected the errors.
- Material and method
"and fishing data such as wind speed and weather were recorded." wind speed etc is not fishing data.
Answer: We have followed your comments, and have corrected the errors.
We fixed the "At the same time, the surface salinity and water temperature were measured, and wind speed and weather were recorded." Line 150-151.
- Material and method
Delete "There are many methods to estimate species richness, such as species number, rarefaction diversity measurement, and various non-parametric estimates [43-45]. These methods can be used to evaluate the integrity of the survey according to the species accumulation curves (SACs) [46] and how rare species are represented in a sample [47], and by comparing the expected theoretical and observed values of species richness.", and " This is example 1 of an equation: ",
Answer: We have followed your comments, and have deleted the paragraphs. The original Line 176-181 and Line 189 are deleted.
- Material and method
"give the reference or version". Line 210.
Answer: We have followed your comments, and have added the reference.
We added the " McAleece, N., Gage, J.D.G., Lambshead, P.J.D., Paterson, G.L.J. (1997) BioDiversity Professional statistics analysis software. Jointly developed by the Scottish Association for Marine Science and the Natural History Museum London." Line 187.
Also see in References 49.
- Results
See pdf file
Answer: We have followed your comments.
- Results
"coma", Line 222.
Answer: We have followed your comments, and have corrected the errors.
- Results
"keep the legend with the table coma", in Table 3, Line 228-229.
Answer: We have followed your comments, and have corrected the errors.
We fixed the "Table 3" in Line 203-204.
- Results
"surveyed" correct "inventoried", in Table 4, Line 228-229.
Answer: We have followed your comments, and have corrected the errors.
We fixed the "Table 4. Number of fish species inventoried at different sampling sites in Minjiang Estuary and adjacent waters. " in Line 231.
- Results
"are" correct "were", always use the past in the paper, Line 246-247.
Answer: We have followed your comments, and have corrected the errors.
We fixed the " The values for spring, summer and winter were 0.259, 0.359 and 0.286, respectively." in Line 219-220.
- Results
"Sob" correct "Sobs" I guess not Sob see method, Line 248.
Answer: We have followed your comments, and have corrected the errors.
We fixed the " Figure 2" in Line 221.
- Results
"delete the space", Line 296.
Answer: We have followed your comments, and have corrected the errors.
We fixed the " Figure 3" in Line 264.
- Results
" SST in full lettre sea surface.... ", Line 297-298.
Answer: We have followed your comments, and have corrected the errors.
We fixed the " Figure 3. The relationship between fish species richness and sea surface temperature (SST) in the Minjiang Estuary and its adjacent waters. " in Line 265-266.
- Results
"Sob" correct "Sobs", Line 299.
Answer: We have followed your comments, and have corrected the errors.
We fixed the " Figure 4" in Line 283.
- Results
"keep the legend with the fig add a point at the end", Line 300-301.
Answer: We have followed your comments, and have corrected the errors.
We fixed the " Figure 4" in Line 284-285.
- Results
Delete "it can be concluded that", Line 311.
Answer: We have followed your comments, and have deleted the errors. The original Line 311 is deleted.
We fixed the "In Figure 5, there is a power exponential function relationship between the fish species richness and the sampling sites (R2 = 0.9820)." in Line 287-288.
- Results
"explain difference between full and empty losange", in Figure 5. Line 292.
Answer: We have followed your comments, and have added the paragraphs.
We fixed the "Curves were fitted using data from all 12 sites (solid dots), and then fish species richness was estimated from 13 to 24 site number (open dots)." in Line 288-290.
- Discussion
See pdf file
Answer: We have followed your comments.
- Discussion
"you can discuss on the interest of other indicators for management (not considered in your study) for societal benefit. By for example adapt to estuary the work done by :
HYPERLINK "https://scholar.google.com/citations?view_op=view_citation&hl=fr&user=wxUPS9IAAAAJ&cstart=20&pagesize=80&citation_for_view=wxUPS9IAAAAJ:hqOjcs7Dif8C" Field investigations and multi‐indicators for shallow water lagoon management: perspective for societal benefit P Brehmer, T Do Chi, T Laugier, F Galgani, F Laloë, AM Darnaude, ...Aquatic Conservation: Marine and Freshwater Ecosystems 21 (7), 728-742", Line 383.
Answer: We have followed your comments, and very much agree with your opinion.
We have added new paragraphs.
We fixed the "Like estuaries, the abiotic and biotic variables of coastal lagoons are highly heterogeneous in space and time, and this heterogeneity complicates the assessment of their ecological status. It is critical to monitor and protect these fragile ecotones and the resources and services they sustain [68]. Bremer et al. [68] emphasized the need for a multifaceted framework to properly assess lagoon conditions and highlighted the need for high-frequency lagoon monitoring to avoid erroneous condition assessments and resulting management plans." in Line 371-377.
We have added anew References 68.
- Discussion
"are" correct "were", and delete "very", Line 385.
Answer: We have followed your comments, and have corrected the errors.
We fixed the "seasonal conditions and species numbers were complete and suitable [40]." in Line 379.
- Discussion
"are" correct "were", Line 392.
Answer: We have followed your comments, and have corrected the errors.
We fixed the " The 153 species of demersal fish collected this time were higher than the research results of two quarters in 2008 and 2009 [37,38]." in Line 386.
- Discussion
"add some references e.g.
Does coastal lagoon habitat quality affect fish growth rate and their recruitment? Insights from fishing and acoustic surveys
P Brehmer, T Laugier, J Kantoussan, F Galgani, D Mouillot
Estuarine, Coastal and Shelf Science 126, 1-6", Line 409.
Answer: We have followed your comments, and have added the references.
We fixed the "69. Brehmer, P.; Do Chi, T.; Laugier, T.; Galgani, F.; Laloë, F.; Darnaude, A.M.; Fian-drino, A.; Mouillot, D. Field investigations and multi-indicators for shallow water la-goon management: perspective for societal benefit. Aquat. Conserv.: Mar. Freshw. 2011, 21 (7), 728–742. http://dx.doi.org/10.1002/aqc.1231
" in References 69.
- Discussion
"you can also propose new sampling technique, I mean complementary one which will allow to complete fishing data in more quantitative way and considering upstream and downstream; e.g.
Amphidromous fish school migration revealed by combining fixed sonar monitoring (horizontal beaming) with fishing data
P Brehmer, T Do Chi, D Mouillot
Journal of Experimental Marine Biology and Ecology 334 (1), 139-150.", Line 436.
Answer: We have followed your comments, and have added the references.
We fixed the "71. Brehmer, P.; Do Chi, T. Mouillot, D. Amphidromous fish school migration revealed by combining fixed sonar monitoring (horizontal beaming) with fishing data. J. Exp. Mar. Biol. Ecol. 2006,334 (1),139-150." in References 71.
- Conclusions
See pdf file
Answer: We have followed your comments.
- Conclusions
"keep one number that you estimate the most representative from you discussion on non parametric method", Line 457.
Answer: We have followed your comments, and have corrected the errors.
We fixed the "According to the number of fish species that appeared in the four cruise surveys, the estimated fish richness was: 250 (Chao 2)." in Line 458-460.
- Conclusions
"why?you mean extinction?", Line 460.
Answer: We have followed your comments.
Although we saw the phenomenon and speculated that “The results of this study reveal the possibility of fish species disappearing.” But we do not have enough evidence to identify it as extinction. So we use the word "disappearing"
We fixed the " The results of this study reveal the possibility of fish species disappearing and not appearing." in Line 462-463.
- Conclusions
"in comparison to what ? the sea?", Line 461.
Answer: We have followed your comments, and have corrected the paragraphs.
We fixed the "It can be seen from the eight reports that the environmental conditions and influencing factors of estuaries are more complex and changeable." in Line 465-467.
- Conclusions
Delete "In short, it is important to study fish richness, especially the impact on endemic species and economically important species, to provide decision-making basis for marine biodiversity conservation.", Line 463-465.
Answer: We have followed your comments, and have deleted the paragraphs. The original Line 463-465, the paragraph is deleted.

Reviewer 2 Report
Dear authors,
Congratulations for your work results.
A clearer separation among authors personal/original data and others data should be revealed more.
In my opinion a following effort in synthethysising the paper will ease the readers effort in reading and understanding it.
The red line of objectives-results-conclusions of the papers can be more highlighted.
The introduction can be more synthetic and relatively general info can be deleted.
The references are not in the proper format.
The citations can be enriched to sustain better the text.
The conclusions can be put in a more accentuated ecological context to raise their value, and consequently to reveal the value of your paper.
Success and all the best.
Reviewer
Author Response
Title: Sustainability Perspective of Minjiang Estuary Coastal Fisheries Management - Estimation of Fish Richness
Reviewer #2: Comments to water-2483437 by Wang et al. (please, also see annotated manuscript):
We are much grateful for your careful reading of our manuscript and your valuable comments and suggestions to help improve the paper. We have now carefully revised the paper in light of all the comments and suggestions. The following is a point-by-point response.
A clearer separation among authors personal/original data and others data should be revealed more.
1. In my opinion a following effort in synthethysising the paper will ease the readers effort in reading and understanding it.
The red line of objectives-results-conclusions of the papers can be more highlighted.
Answer: We have followed your comments, and very much agree with your opinion.
Same as reviewer #1 comments 1-6.
We have revised the Abstract. Line 11-26.
2. The introduction can be more synthetic and relatively general info can be deleted.
Answer: We have followed your comments, and very much agree with your opinion.
Same as reviewer #1 comments 7.
We have revised the Introduction. Line 30-101.
3. The references are not in the proper format.
Answer: We have followed your comments, and have corrected the errors.
We have revised the References. Line 492-642.
4. The citations can be enriched to sustain better the text.
Answer: We have followed your comments, and very much agree with your opinion.
We have added six new citations.
5. The conclusions can be put in a more accentuated ecological context to raise their value, and consequently to reveal the value of your paper.
Answer: We have followed your comments, and very much agree with your opinion.
We have added new paragraphs.
We fixed the "The estuary is affected by many factors such as river freshwater erosion, ocean tides, waves, currents, etc. The physical and chemical conditions are complex and the environmental factors are changeable. Scientifically ascertaining the number of biological species is the key and prerequisite for biodiversity research. The comprehensive reliability of species richness estimation studies in the Minjiang Estuary has a decisive impact on improving conservation practices, formulating management policies, and protecting fishery resources." in Line 451-457.
